# Using Restaurant POI Data to Explore Regional Structure of Food Culture Based on Cuisine Preference

**Shangjing Jiang** [1,2,3] ⓘ**, Haiping Zhang** [1,2,3,*] ⓘ**, Haoran Wang** [1,2,3] ⓘ**, Lei Zhou** [4] **and Guoan Tang** [1,2,3]

1   School of Geographic Science, Nanjing Normal University, Nanjing 210023, China;
    181302072@njnu.edu.cn (S.J.); 191302046@njnu.edu.cn (H.W.); tangguoan@njnu.edu.cn (G.T.)
2   Key laboratory of Virtual Geographic Environment, Ministry of Education, Nanjing Normal University,
    Nanjing 210023, China
3   Jiangsu Center for Collaborative Innovation in Geographical Information Resource Development and
    Application, Nanjing Normal University, Nanjing 210023, China
4   School of Geographic and Biologic Information, Nanjing University of Posts and Telecommunications,
    Nanjing 210023, China; zhoulei@njupt.edu.cn
*   Correspondence: 90775@njnu.edu.cn

**Abstract:** As a result of the influence of geographical environment and historical heritage, food preference has significant regional differentiation characteristics. However, the spatial structure of food culture represented by the cuisine culture at the regional level has not yet been explored from the perspective of geography. Cultural regionalization is an important way to analyze and understand the spatial structure of food culture. It is of great significance to deeply mine intra-regional homogeneity and scientifically cognize inter-regional cultural characteristics. This study aims to explore such patterns by focusing on the restaurants of the eight most famous cuisines in Mainland China. Initially, the density based geospatial hotspot detector method is proposed to analyze and mapping the spatial quantitative characteristics of the eight major cuisines. A heuristic method for geographical regionalization based on machine learning was used to analyze spatial distribution patterns in accordance with the proportion of these cuisines in each prefecture-level city. Results show that some types of single-category cuisines have a stronger spatial concentration effect in the present, whereas others have a strong diffusion trend. In the comprehensive analysis of multicategory cuisines, the eight major cuisines formed a new structure of geographical regionalization of Chinese cuisine culture. This study is helpful to understand regional structure characteristics of food preference, and the density-based hotspot detector proposed in this paper can also be used in the analysis of other type of point of interest (POI) data.

**Keywords:** food culture; cultural regionalization; Chinese cuisines; machine learning; spatial pattern

## 1. Introduction

Diet is not only a basic material element to meet human physiological needs, but it is also an important carrier of human cultural elements. The cultural phenomena related to the demand, production, and consumption of food were investigated in food culture studies [1,2]. Food culture has the characteristics of multi-disciplinary research, and the study of food culture from the perspective of geography emphasizes the human–environment relationship [3] and the geographical differentiation law reflected by the preference of food culture [4,5]. Geographical regionalization and mapping of food culture are important analytical and presentation methods to reveal regional differences of food culture, which are significant for mining local food culture resources and understanding local food culture characteristics [6]. At the same time, the regional structural characteristics presented by the food culture regionalization provide references for studying human–environment relationship in the geography of food culture [7].

Research on regional food culture is a trending topic in geography and related disciplines. Research topic includes two aspects, namely, what influences regional food

cultures and what regional food cultures have influence on. The former focuses on how economy, politics, history, religion, and geographical environment affect regional food culture, whereas the latter focuses on the impact of regional food culture on people's health, tourism, and other aspects, namely, impact of regional food cultures on the formation of sense of place and the construction of local cultural brands and symbols. In terms of the influence of economy on food culture, certain scholars reported that fast food chains in developed regions will be the first to enter and then affect the local food culture [8,9]. Several researchers also claimed that groups with different incomes will produce various food consumption cultures [10]. In the research on the relationship between regional food culture and politics, Tellstrom et al. studied how governments interpret and present their national image by shaping local food culture [11]; that is, how the differences in political culture affect the food culture and daily food consumption patterns of people in different regions [12,13]. Historical and cultural customs and population migration in the history also profoundly impact local food cultures [14,15]. The relationship between religions and regional food culture is also a hot topic; for example, religious beliefs affect structures of food cultures, and the degree of attention to the green food differs between religious and nonreligious people [15–17].

In studies on geographical environment and food culture, climate, food materials and their proximity to coastal areas are the core influencing factors for the formation of regional food culture [18–20]. These studies discussed food culture as affected objects, that is, which factors affect the production and reproduction of food cultures. In the studies of food cultures, the relationship between regional food culture and health attracted the attention of scholars as the major influencing factor. For example, certain studies suggested that the food cultures of different regions always tend to develop in the direction that is beneficial to people's health. The food cultures of tropical regions help people keep cool, whereas those of cold regions help people stay warm [21,22]. Certain studies also suggested that the existing food culture in certain places is inconducive to people's health [23,24]. Another widely followed research topic is how regional food culture affects the development of tourism. Several studies reported that people's pursuit of local characteristic cuisine stimulated the interest of tourism developers and tourists in food tourism. In terms of academic research, food tourism is also an expanding field [25]. For example, Robinson et al. conducted a data-driven empirical study on gastronomic tourism among tourists [26]. In recent years, several scholars also focused on the production of cross-regional food culture and the construction of local sense [27,28].

Most studies regarded food as a culture and emphasized regional characteristics as inherent geographical attributes of food cultures. These studies also highlighted the human–environment relationship reflected in food cultures. However, the regional characteristics of people's food preference differences emphasized by geography are rarely mentioned in the literature. Only several studies are available on the establishment of boundaries of food culture and homogeneous-culture regions by geographical regionalization method when the same region involves multiple food cultures. China puts importance on food culture. However, it has vast territory, diverse food culture, and evident regional differences in food culture. Hence, conducting geographical regionalization based on various food cultures in the Chinese context is necessary to have a deep understanding of the characteristics of regional food cultures.

Cultural regionalization is the basis for deeply mining regional cultural characteristics and scientifically cognizing regional differences in culture [29]. It is an important way to extract and understand cultural regions. Cultural region is one of five main themes in traditional cultural geography research and the others are cultural landscape, ecology, diffusion and integration [30]. As an important cultural element in China, the food culture represented by Chinese cuisines has a profound influence on the regional structure of Chinese culture [30,31]. Existing researches are mainly based on cultural elements to extract culture regions, such as dialects and history [32]. However, regional culture is a process of dynamic change. Some cultural phenomena show a long-term slow and

gentle dynamic process of change (such as dialects and history), while others show a faster and sharper process of dynamic change [33–35]. Food represents a cultural element of rapid change [36,37]. This paper is based on the study of food culture regionalization of restaurant POI data which owns nearly the whole restaurant samples in China. It is of great significance for understanding the regional characteristics of food culture and enhancing the identity of regional culture [29].

In terms of research methods, geographical regionalization is usually adopted in the following ways: semi-manual cartography method based on experience and data [38,39], traditional GIS overlay analysis and cartographic synthesis method [40,41], and clustering method based on machine learning [42,43]. The semi-manual cartography method has low quantification degree and large workload. Although methods based on overlay analysis and cartographic synthesis have a high degree of automation, handling the weight distribution of geographical objects with complex features is difficult. At present, the multisource clustering method based on machine learning has been introduced into the research of geographical regionalization and extended into the multisource clustering method with spatial constraint. In addition to its ability to handle the complex geographical regionalization with multi-source characteristics, the main advantage of this method is that it is a clustering method based on object model. The multisource clustering method with spatial constraints is adopted to conduct geographical regionalization of food cultures with multiple characteristics, which not only give play to the advantages of the method but also solve two problems, namely, the complex features of cultural phenomenon and the difficulty of using traditional method for a reasonable regionalization result.

Density based hotspot detection is another appropriate way to discover the spatial quantitative characteristics due to it can identify the spatial distribution pattern of objects or elements [44]. By the way of hotspot detection, some researchers have studied on spatial patterns of points with different semantics. Typical hotspot detecting algorithms include DBSCAN, OPTICS, Kernel Density Estimation (KDE), etc. The analysis results of DBSCAN and OPTICS are density-connected objects and tell us which objects with high density [45–47]. By contrast, the analysis result of KDE is local maximums based on the density field, thus tell us which area with high density [43]. However, these methods are difficult to extract the specific locations with the local maximum density, this paper proposes a density-based hot spot detection method to achieve this goal, so that better represents spatial quantitative characteristics of restaurants.

This study takes the entire China as research area, adopts the restaurants of the eight most famous great cuisines as the main data source, and takes prefecture-level cities as the main research units. It uses the preference index of each prefecture-level cities for each cuisine as the regional analysis variable and adopts regionalization method with spatial constraints for geographical regionalization of food cultures. The principle is to maintain the similarity of food preference structure within the same area and maximize the difference in different regions. Finally, China is divided into several continuous food culture regions. Thus, the quantification of the regional structure of Chinese food culture based on the eight traditional cuisines is conducive to deep cognition of the spatial differentiation rules of Chinese food culture and excavation of local characteristic food culture.

## 2. Study Area and Data Description

The cultural background of Chinese food is different from that of Western countries such as Europe and the United States. The majority of Chinese people's diet is mainly Chinese food. Although Chinese food is made up of various regional cuisines, it includes not only the eight major cuisines. After a long period of evolution and its own system, the eight major cuisines with distinctive local flavored characteristics, are widely recognized by the society and the most influential local cuisines in China. In China, food preferences vary greatly from region to region, and have always been seen as a cultural symbol which is used to distinguish cultural differences between people in different regions. This provides rationality for treating food preferences as a cultural phenomenon, as well as the possibility

of regionalization based on food culture. From the perspective of cultural influence, the eight major cuisines can be regarded as the representation of food preference. The concentration of restaurants with eight major cuisines indicates the concentration of their audiences, and the proportion of different cuisines in the areas indicates the composition of food preferences in the areas. Thus, Clustering and regionalization based on eight major cuisines can represent the regional structure of food culture preference.

The research area is Mainland China. Taiwan, Hong Kong, and Macao are excluded due to the lack of data on catering facilities in these regions. The research area comprises 31 secondary administrative regions (provinces, autonomous regions, and municipalities). In the regional classification and geographical regionalization of food cultures, the basic geographical unit adopted in this study is prefecture-level city, which is the third administrative regionalization after the province or autonomous region. Prefecture-level cities are adopted as the analysis unit because their population reached a certain scale, and the number of catering service facilities can guarantee the diversity to a certain extent to avoid the situation where the number of certain cuisines is zero in certain analysis units. Assuming that lower-level county is taken as the analysis unit, not all counties would have restaurants with the eight major cuisines. Because there are 2844 county-level regions in the country, and the total number of restaurants with certain types of the eight major cuisines is not larger than 5000. The whole research area is shown in Figure 1. The yellow highlights in Figure 1 are the provinces where the eight major cuisines are located and the birthplace of their respective cuisine culture.

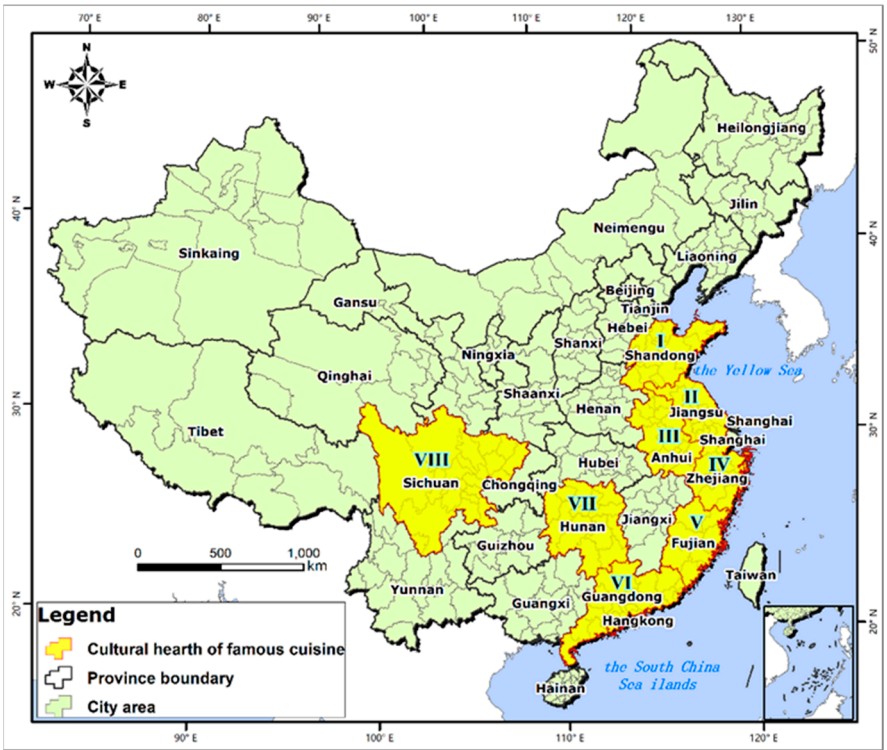

**Figure 1.** Study area and cultural hearth of eight famous Chinese cuisines.

As the representative of Chinese food culture, the eight major cuisines have a long history of inheritance. These specialty restaurants can now be found in most Chinese cities, with some expanding rapidly and others slowly. For example, Sichuan and Hunan cuisines have the fastest diffusion, whereas Jiangsu and Shandong cuisines have the slowest (Table 1). The data was extracted from the online map service of AMap in 2018, China's most famous online map service provider. In AMap map, the POI of catering type contains the classification field of specific catering type, which contains the classification value of traditional eight cuisines. Eight major cuisines and their geographical location information

can be extracted using this field. Table 1 lists the names of the eight major cuisines, major origins, total number of restaurants nationwide, and percentage of restaurants in each cuisine.

**Table 1.** Quantity and proportion of eight famous Chinese cuisines.

| Cuisine Categories | Major Origin Place | Serial Number | Amount | Proportion |
|---|---|---|---|---|
| Shandong cuisine | Shandong | I | 4996 | 2.00% |
| Jiangsu cuisine | Jiangsu | II | 2273 | 1.01% |
| Anhui cuisine | Anhui | III | 9706 | 4.32% |
| Zhejiang cuisine | Zhejiang | IV | 18,310 | 8.15% |
| Fujian cuisine | Fujian | V | 1495 | 0.67% |
| Guangdong cuisine | Guangdong | VI | 16,771 | 7.46% |
| Hunan cuisine | Hunan | VII | 41,585 | 18.50% |
| Sichuan cuisine | Sichuan | VIII | 129,650 | 57.68% |

## 3. Research Framework and Methodology

### 3.1. Research Framework

The whole research framework of this study is shown in Figure 2. Initially, restaurants of eight cuisines are extracted from 7.5 million restaurants in China (classified labels in POI data of each catering service are provided). The analysis is then performed from the aspects of quantity and ratio. At the level of quantitative features, the hotspot detector is used to detect the spatial hotspots of Chinese restaurants of eight cuisines. The natural breaks method is then used to grade the hotspots detected. Finally, the spatial quantitative distribution characteristics of various cuisines are analyzed on the basis of hotspots with hierarchical structure. In, restaurants of various cuisines tend to concentrate in cities with large and high-density population and developed economy. To eliminate this effect, the proportion of the number of restaurants corresponding to each cuisine in the total restaurants in the prefecture-level city is calculated. At the ratio level, the method in Section 3.2.2 is adopted to classify all prefecture-level cities by taking the ratio of eight cuisines in each prefecture-level city as a regional variable. The result of type regionalization can reflect the structural characteristics of eight cuisines in each prefecture-level city and identify which eight cuisines are homogeneous in ratio structure. If a homogeneous prefecture-level city has spatial agglomeration, then comprehensive regionalization can be conducted on the basis of the ratio of eight cuisines as a regional variable to obtain a zoning map reflecting the regionalization and differentiation mode.

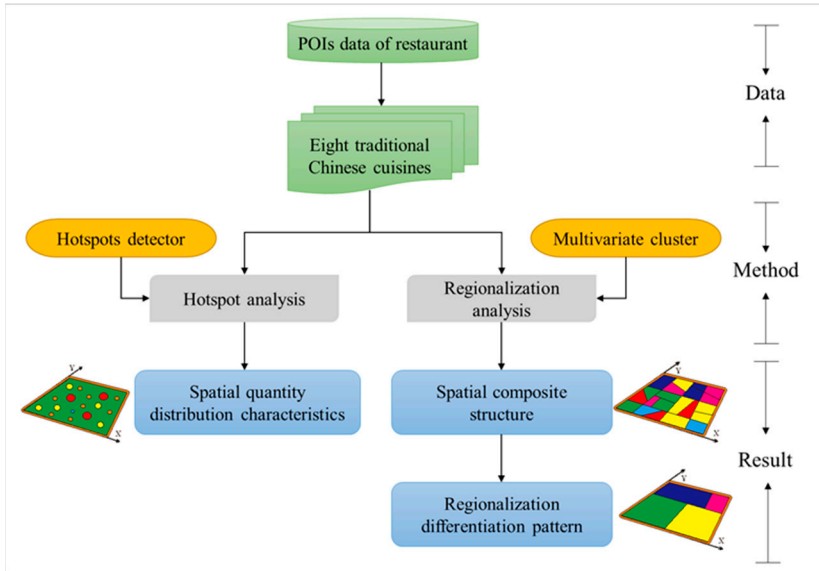

**Figure 2.** Research framework (analysis data, method, and result).

*3.2. Methodology*

3.2.1. Density-Based Hotspot Detecting Method

An analytical method that can detect the maximum local density of all restaurants in the spatial distribution, where these local maximum values are called hotspots, is constructed to detect the maximum local density of restaurants. Hotspots can be classified with the clustering algorithm because of different sizes in the local maximum values. The hotspot ranking structure in the whole analysis region can then be obtained. A probability density surface is constructed via kernel density method for each cuisine. Among many kernel functions, the most commonly used function is adopted, namely the Gaussian kernel function. The calculation formula is as [31]

$$f(x,y) = \frac{3}{n\pi r^2} \sum_{i=1}^{n} E\left[1 - \frac{(x - x_i)^2 + (y - y_i)^2}{r^2}\right]^2 \quad (1)$$

where $f(x,y)$ denotes the probability density of calculation unit, $r$ denotes the bandwidth, $n$ denotes the number of sample points in the bandwidth range, $x_i$ and $y_i$ are coordinates of sample point, $x$ and $y$ are coordinates of calculation unit, $E$ denotes the kernel function.

The obtained probability density is Matrix A with m rows and n columns, which can be expressed as

$$A = \left(a_{ij}\right)_{mn} = \begin{bmatrix} a_{11} & a_{12} & \cdots & a_{1n} \\ a_{21} & a_{22} & \cdots & a_{2n} \\ \cdots & \cdots & \cdots & \cdots \\ a_{m1} & a_{m2} & \cdots & a_{mn} \end{bmatrix} (i = 1, 2, \ldots, m; j = 1, 2, \ldots, n.) \quad (2)$$

For any element $a_{ij}$ in Matrix A, the eight adjacent neighborhood elements, which constitute A matrix subset $W_{ij}$, are obtained, and $W_{ij} \in A$ is satisfied. The element, where the maximum value in $W_{ij}$, is obtained. All the element values in $W_{ij}$ are set as the maximum value, and the new matrix subset $W_{ijmax}$ is finally obtained. These steps are performed for all the elements in matrix A to obtain a new matrix B. For good understanding, the above logic is illustrated with an example. Here, the sample matrix is expressed as

$$A' = \left(a_{ij}\right)_{44} = \begin{bmatrix} a_{11} & a_{12} & a_{13} & \mathbf{a_{14}} \\ a_{21} & \mathbf{a_{22}} & a_{23} & a_{24} \\ a_{31} & a_{32} & a_{33} & a_{34} \\ \mathbf{a_{41}} & a_{42} & \mathbf{a_{43}} & a_{44} \end{bmatrix} (\mathbf{a_{14} > a_{22} > a_{41} > a_{43}}) \quad (3)$$

The elements in bold are assumed to be locally maximized.

$$W_{11} = \begin{bmatrix} 0 & 0 & 0 \\ 0 & a_{11} & a_{12} \\ 0 & a_{21} & \mathbf{a_{22}} \end{bmatrix} \quad (4)$$

Certain neighboring elements do not exist because $a_{11}$ is at the edge of the matrix. Hence, it is zero. $W_{11max}$ is obtained and satisfied as

$$W_{11max} = \begin{bmatrix} 0 & 0 & 0 \\ 0 & a_{22} & a_{22} \\ 0 & a_{22} & a_{22} \end{bmatrix} \quad (5)$$

Similarly,

$$W_{12max} = \begin{bmatrix} 0 & 0 & 0 \\ a_{22} & a_{22} & a_{22} \\ a_{22} & a_{22} & a_{22} \end{bmatrix}, W_{13max} = \begin{bmatrix} 0 & 0 & 0 \\ a_{14} & a_{14} & a_{14} \\ a_{14} & a_{14} & a_{14} \end{bmatrix}, W_{14max} = \begin{bmatrix} 0 & 0 & 0 \\ a_{14} & a_{14} & 0 \\ a_{14} & a_{14} & 0 \end{bmatrix} \quad (6)$$

For an element without boundary effect, such as $a_{23}$, because of

$$a_{23} = \begin{bmatrix} a_{12} & a_{13} & a_{14} \\ a_{22} & a_{23} & a_{24} \\ a_{31} & a_{32} & a_{34} \end{bmatrix} (a_{14} > a_{22}) \tag{7}$$

The calculation result of $W_{23max}$ is

$$W_{23max} = \begin{bmatrix} a_{14} & a_{14} & a_{14} \\ a_{14} & a_{14} & a_{14} \\ a_{14} & a_{14} & a_{14} \end{bmatrix} \tag{8}$$

When all the elements are computed, the local maximum matrix $B'$ of the entire example matrix $A'$ is computed as follows:

$$B' = \begin{bmatrix} a_{22} & a_{22} & a_{14} & a_{14} \\ a_{22} & a_{22} & a_{22} & a_{43} \\ a_{41} & a_{41} & a_{22} & a_{43} \\ a_{41} & a_{41} & a_{43} & a_{43} \end{bmatrix} \tag{9}$$

Finally, the difference value between the local maximum matrix $B'$ and the original probability density matrix $A'$ is calculated, and each element is reassigned as follows: when $a_{ij} < 0$, $a_{ij} = 0$; when $a_{ij} \geq 0$, $a_{ij}=1$. Then another new matrix $C'$ can be obtained as follows:

$$C' = B' - A' = \begin{bmatrix} 0 & 0 & 0 & 1 \\ 0 & 1 & 0 & 0 \\ 0 & 0 & 0 & 0 \\ 1 & 0 & 1 & 0 \end{bmatrix} \tag{10}$$

In $C'$, elements with a value of 1 are at the hotspot. If $C'$ is multiplied to the prime, then the matrix $R'$ with the hotspot value is obtained.

$$R' = A'C' = A'(B' - A') = \begin{bmatrix} 0 & a_{12} & 0 & a_{14} \\ 0 & a_{22} & 0 & 0 \\ 0 & 0 & 0 & 0 \\ a_{41} & 0 & a_{43} & 0 \end{bmatrix} \tag{11}$$

The method of detecting hotspots based on a restaurant service facility is shown above, and the results are obtained by detecting the hotspots of each cuisine of restaurants and be divided into different levels by natural break clustering method.

### 3.2.2. Regional Preference Index of Cuisine

Regional preference index refers to the popularity of a cuisine in a certain region. In this study, $R_i$ represents the $i$th geographical unit; $N_i$ represents the total number of restaurants in the geographic unit $R_i$; and $M_{ij}$ represents the number of restaurants in cuisine $j$ in the geographical unit $i$. Regional preference index can then be calculated by

$$RPI_{ij} = \frac{M_{ij}}{N_i} \tag{12}$$

In this study, the value of $j$ is $j \in (1, 2, 3, 4, 5, 6, 7, 8)$, and each element represents every cuisine. The geographical unit is prefecture-level city. Shanghai is the geographical unit $i(i = 1)$, whereas Sichuan is the cuisine $j(j = 1)$. In this study, $N_1 = 18,000$, and $M_{11} = 4000$. Hence, $RPI_{11} = \frac{400}{18,000} = 0.011$. On this basis, the preference index of Sichuan cuisine is 0.011 in Shanghai. Similarly, we can calculate the other preference index of the other cuisines in Shanghai and calculate their preference index using the same methods.

### 3.2.3. Comprehensive Regionalization Method

Geographic regionalization is a process for classification of spatial geographical objects, and the result is the merging of adjacent regions with homogeneity in multiple attributes into the same geographical region in the form of surface elements to form a regionalized map composed of multiple regions. This process is significant to understand the similarity and heterogeneity between regions deeply. In this study, a heuristic method of regionalization in machine learning that can effectively realize the classification of regions is adopted. The classification without region constraint only needs K clustering method based on multiple attribute information. The methods with region constraint include the construction of adjacency matrix, generation of minimum span tree based on adjacency matrix, and segmentation of minimum span tree. The relationship between regional classification without regional constraints and regional integration with regional constraints can be expressed as follows: regional integration is meaningful only when the results of regional classification have significant homogeneous regional agglomeration effect. Therefore, we classify types using this method and judge the clustering degree of homogeneous regions. If the clustering effect is evident, then the geographic regionalization will be further developed. The regionalization process is briefly introduced below.

A set of more than 300 local surface sources in China is defined as O. The surface source set O contains the attribute set $X = \{A_1, A_2, \ldots, A_n\}$, which in turn contains the vector x = $\{a_1, a_2, \ldots, a_n\}$. The attribute values in x correspond to the attribute columns in the attribute set X. If the region constraint is disregarded and only the type regionalization is performed, then the region classification in O is implemented directly via K clustering. If the region constraint is considered, then the adjacency matrix among surface sources in O needs to be constructed. The adjacency topological relation between surfaces in set O can be expressed as G = (V, L), where V and L are the set of nodes and edges of the topological tree, respectively. For the two nodes $v_i$ and $v_j$ with adjacent relations, the attribute vectors of the plane object in O corresponding to them are $x_i$ and $x_j$, respectively. The cost distance between the objects represented by $v_i$ and $v_j$ is defined as $d(v_i, v_j)$, and the formula is as follows [40]:

$$d(v_i, v_j) = \sum_{l=1}^{n} \left( x_{il} - x_{jl} \right)^2 \tag{13}$$

where $x_{il}$ and $x_{jl}$ are the attribute of node $v_i$ and $v_j$, respectively. The sum of all the differences between the attributes of the two nodes is the cost distance defined in this article. The whole tree G is divided into c node clusters $G_1, G_2, \ldots, G_c$ by using the construction and segmentation strategy of the minimum span tree.

This method also has the perplexity of NP problem because it belongs to the unsupervised space classification algorithm. Therefore, no optimal solution exists. The number of types or regions can be customized, or the validity of the number of different categories can be determined by the Calinski–Harabasz pseudo-F test. Calinski–Harabasz pseudo-F statistic provides the optimal classification scheme by calculating the ratio $d$ of inter- and intra-cluster variance. In other words, it is the ratio reflecting intragroup similarity and intergroup difference. The evaluation formula is as follows [48]:

$$F = \frac{\left( \frac{R^2}{n_c - 1} \right)}{\left( \frac{1 - R^2}{n - n_c} \right)} \tag{14}$$

whereas:

$$R^2 = \frac{SST - SSE}{SST} \tag{15}$$

where SST reflects the differences between different groups, whereas SSE reflects the similarity of attributes within a group. The formula for the two variables can be expressed as

$$SST = \sum_{i=1}^{n_c} \sum_{j=1}^{n_i} \sum_{k=1}^{n_v} \left( v_{ij}^k - \overline{v^k} \right)^2 \tag{16}$$

$$SSE = \sum_{i=1}^{n_c} \sum_{j=1}^{n_i} \sum_{k=1}^{n_v} \left( v_{ij}^k - \overline{v_t^k} \right)^2 \tag{17}$$

where n is the number of surface elements; $n_i$ is the number of elements in the first group I; $n_c$ is total number of regions; $n_v$ is variables involved in the regionalization; $v_{ij}^k$ represents the value of variable k of the key element j in the group I; $\overline{v^k}$ is the mean value of variable k; and $\overline{v_t^k}$ is the mean value of variable *k* in group *i*. In this study, Calinski–Harabasz pseudo-F test is used to evaluate the optimal group number for regionalization. When F test value reaches the maximum value at a certain classification value, this classification is the best grouping value with the smallest difference within the group and the largest difference between groups.

## 4. Result

### 4.1. Spatial Quantitative Characteristics

The first-level hotspots of Shandong and Jiangsu cuisines were located in their originally cultural region, whereas the second- and third-level hotspots were relatively fewer (Figure 3a,b). The diffusion trend of Shandong cuisine was insignificant, but the Jiangsu cuisine diffused to the north. The hotspots of Shandong cuisine covered the whole country. Conversely, no fifth-level hotspots of Jiangsu cuisine existed. In the distributing map of hotspots of Anhui cuisine, the only hotspot of the first level was located in Shanghai city outside the originally cultural region (Figure 3c). Hunan cuisine also showed a similar phenomenon (Figure 3g). The only first-level hotspot of Hunan cuisine was located in Guangdong Province rather than in its own originally cultural region. Anhui, Shandong, and Jiangsu cuisines had the same features wherein the hotspots in the higher three levels were fewer than in other levels. In addition, Anhui cuisine also had a significant trend of diffusion to the north. By contrast, the middle and high-level hotspots of Zhejiang and Fujian cuisines were widely distributed throughout the middle and east of China (Figure 3d,h). The Zhejiang cuisine diffused significantly to the north, especially along the coast (Figure 3d). Fujian cuisine formed three second-level hotspots and two first-level hotspots within the province and formed a second-level hotspot in Guangdong and Shanghai respectively, which were economically developed areas. For Guangdong cuisine, only one first-level hotspot was located in the originally cultural region, and only one second-level hotspot was located in Shanghai. The most popular cuisines were Hunan and Sichuan cuisines among eight major restaurants. Sichuan cuisine had the most first- and second-level hotspots. These hotspots were distributed widely all over the country. Hunan cuisine had less hotspots than Sichuan cuisine (Figure 3g,h).

The spatial diffusion ability of northern cuisines (such as Shandong or Jiangsu cuisines; Figure 3a,b) was weak, whereas southern cuisines had strong spatial diffusion ability (Figure 3g). Shandong cuisine had the weakest diffusion ability among the eight major cuisines, whereas Sichuan cuisine had the strongest diffusion ability. Among the eight major cuisine provinces, Anhui Province was the weaker one in the north, and Hunan Province was the weaker one in the south. The cuisine of these two provinces had only one first-level hotspot and was located outside the originally cultural region. The first-level hotspot of Anhui cuisine was located in Shanghai, which was the most developed city in its neighborhood. Meanwhile, the first-level hotspot of Hunan cuisine was located in Guangzhou, which was the most developed city in its neighborhood. In short, the spatial distribution of quantity and the characteristics of hierarchical structure varied greatly among different cuisines.

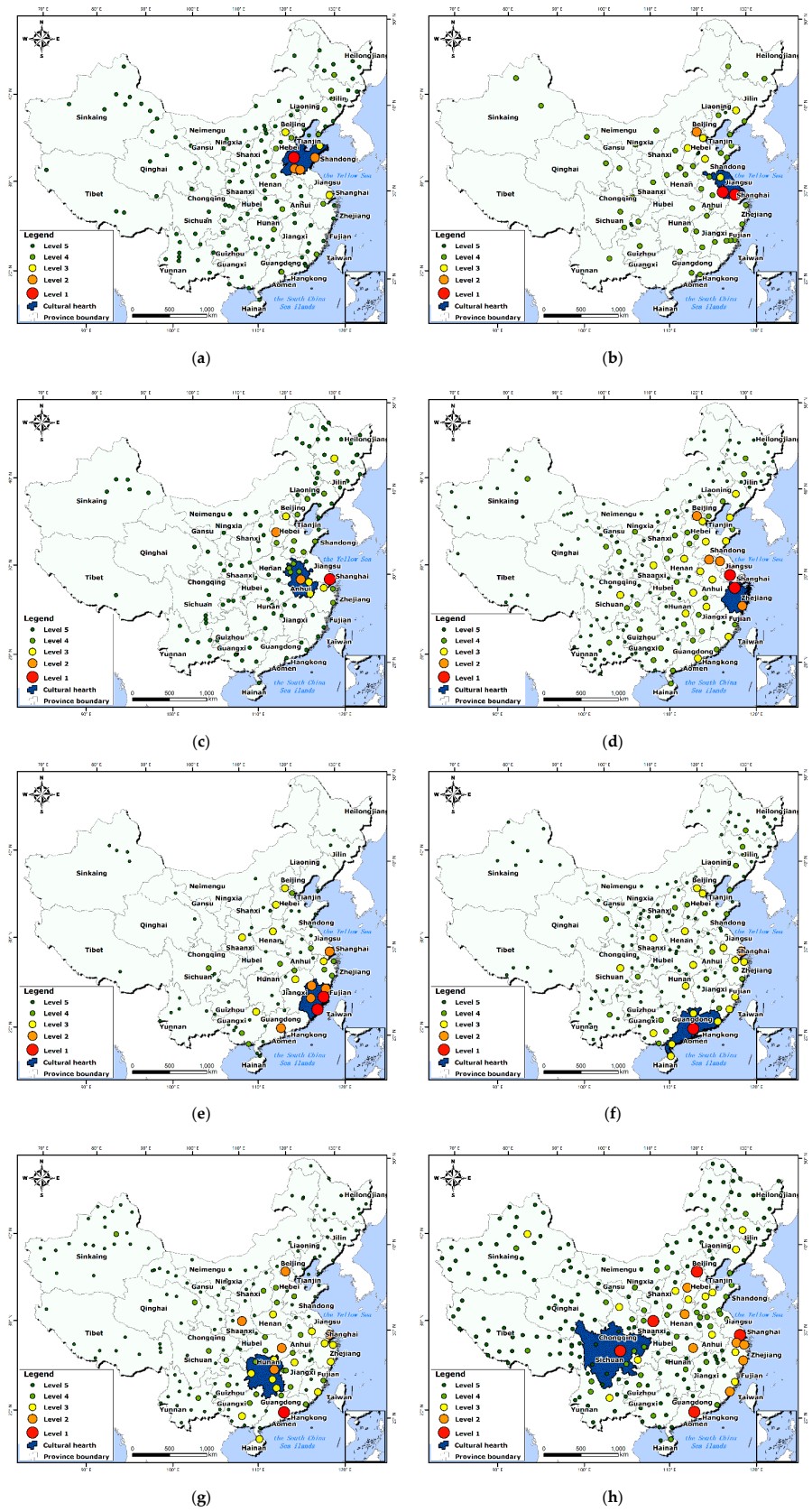

**Figure 3.** Results of hotspots detecting and divide rank of Eight Chinese cuisine, (**a**) Shandong cuisine, (**b**) Jiangsu cuisine, (**c**) Anhui cuisine, (**d**) Zhejiang cuisine, (**e**) Fujian cuisine, (**f**) Guangdong cuisine, (**g**) Hunan cuisine, (**h**) Sichuan cuisine respectively.

### 4.2. Regionalization Differentiation Structure

From a regional perspective, exploring the spatial distribution of the proportion of eight major cuisines will help further exploration of people's preferences in different regions for various cuisines. The ratio of restaurants corresponding to different cuisines in each prefecture-level city to the total number of restaurants in that prefecture-level city was calculated. The method introduced in Section 3.2.3 was used to model all prefecture-level cities. The pseudo-F value was the highest when the number of classifications was 20. Hence, the threshold of the number of classifications was set to 20. The result of region type regionalization was obtained (Figure 4). To ensure the integrity of the analysis results, we classified Hong Kong, Macau, and Taiwan as the same types as their closest geographical neighbors, and the same strategy was applied to regionalization. The prefecture-level cities with Type 1 were widely distributed all over the country with the overall distribution trend from northeast to southwest, and the food preferences of the eight major cuisines in these prefecture-level cities were similar. In addition, many prefecture-level cities with the same classification number showed spatial agglomeration effect, which means that the local cuisine preferences were similar. This phenomenon occurs especially in the densely populated and economically developed eastern and southern coastal areas. For example, prefecture-level cities with Type 20 were primarily concentrated in Shandong Peninsula, and prefecture-level cities with type code 18 were primarily concentrated in Fujian Province. The ubiquitous existence of this spatial autocorrelation phenomenon created a prerequisite for the synthesis and geographic analysis of homogeneous regions.

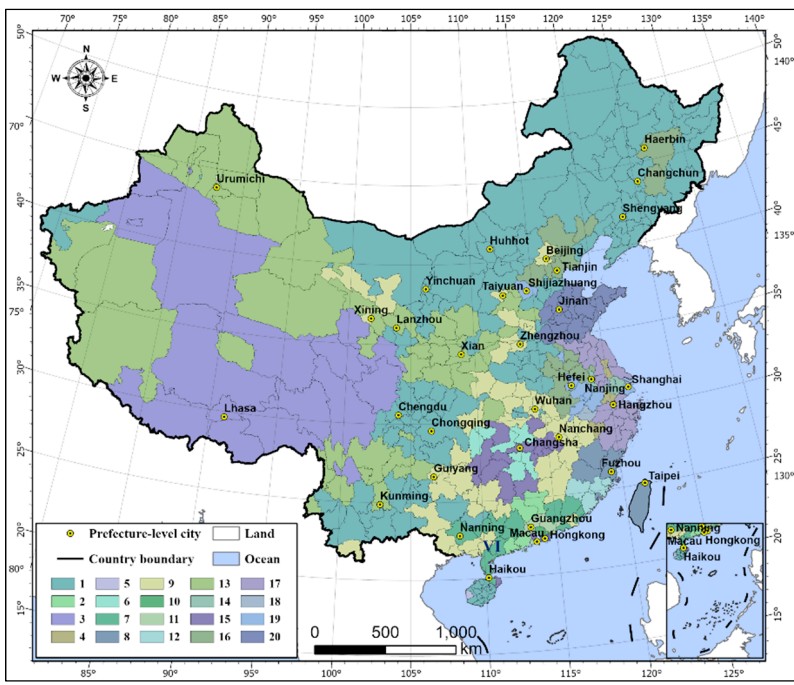

**Figure 4.** The classification result of prefectures based on cuisine preference.

The ratio of restaurants to the total number of restaurants in each city served as regionalized variables. The results were analyzed by geographic regionalization method with spatial constraints in Section 3.2 (Figure 5). The number of regions corresponding to the maximum value obtained by pseudo-F test was 8. Thus, the threshold 8 was used as the number threshold of geographical regions. The Roman numeral I–VIII was used to number eight regions. From the overall structure of the regionalization results, the thinner the coastal area was, the closer the inland area was, the coarser the partition size was. For example, the whole northern region was divided into two parts, namely, northwest (VIII) and northeast (I). In turn, the coastal area was divided into four regions, namely, Region II, III, V, and VI. The southwest area was divided into two major regions, namely, Region IV

and VII. These cultural regions reflected the structural characteristics of the popularity of the eight cuisines in the same cultural area and the homogeneity of the development of these cuisines in these areas. The spatial differentiation structure of the eight major cuisine cultures were reflected among regions.

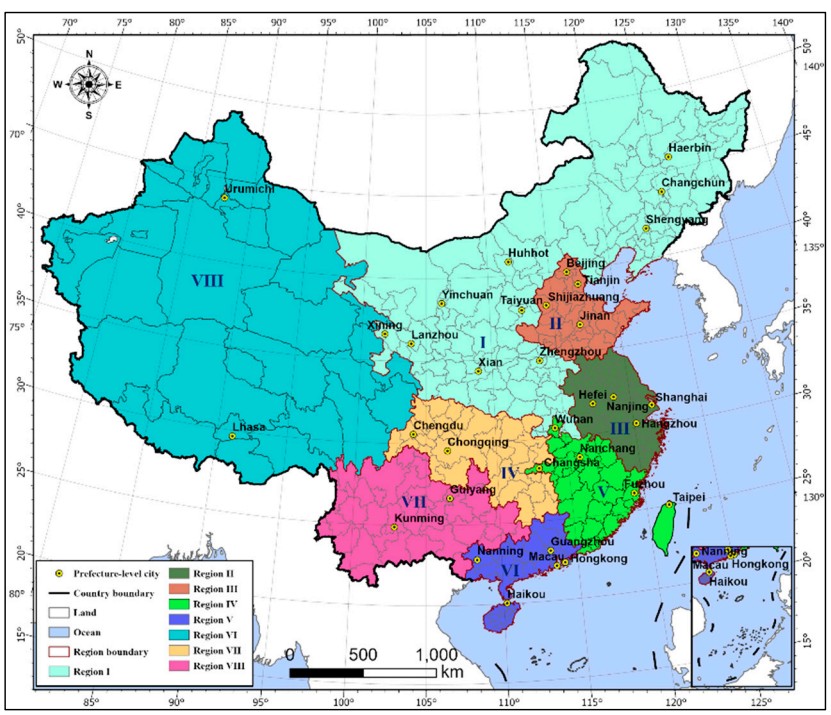

**Figure 5.** Regionalization result of food culture based on cuisine preference.

To illustrate the structural characteristics of the proportion these cuisines in each district further, the ratio boxes and their mean lines of each kind of cuisine at prefecture level and city level were drawn, as shown in Figure 6 and Table 2. The proportion of restaurants corresponding to Sichuan, Fujian, Shandong, and Zhejiang cuisines was relatively stable in more than 300 prefecture-level cities, whereas the proportion of other cuisines in different prefecture-level cities was relatively unstable. The eight regions can be divided into two categories, Region II to VII with high standardized values, while Region I and VIII with low standardized values. Region II had a strong preference for Zhejiang, Jiangsu and Anhui cuisines. In addition, Region II had a high preference, which was close to the third quartile threshold, for Guangdong cuisine. Other cuisines were on average or low level. In Region III, Shandong and Anhui cuisines were the main preferences, whereas the preferences for other cuisines were at a moderate level. Region IV had the strongest inclusiveness to all major cuisines, with a high preference for Fujian, Guangdong, Hunan, and Zhejiang cuisines. The preferred degree of zoning V to Guangdong cuisine, Hunan cuisine was relatively high, Shandong cuisine, Zhejiang cuisine was relatively low, and the preferred degree to other cuisine was in the middle level. Region VI had a high preference for Sichuan, Jiangsu, and Guangdong cuisines, a low preference for Anhui cuisine, and a medium preference for other cuisines. Region VII, which basically belonged to Hunan cuisine's catering culture area, had a strong preference for Hunan cuisine. Combining with the mean line, the proportion of characteristic restaurants of these cuisines in Region VIII remained at a low level. On this basis, people in this area had a neutral preference for eight cuisines and had no particular preference of any cuisine. Region I had a low preference for Anhui cuisine and a medium preference for other cuisines. Generally, Anhui, Guangdong, Hunan, and Zhejiang cuisines were popular in many districts. Particularly, Hunan and Sichuan cuisines, which were above moderately preferred in almost every district, were the most popular.

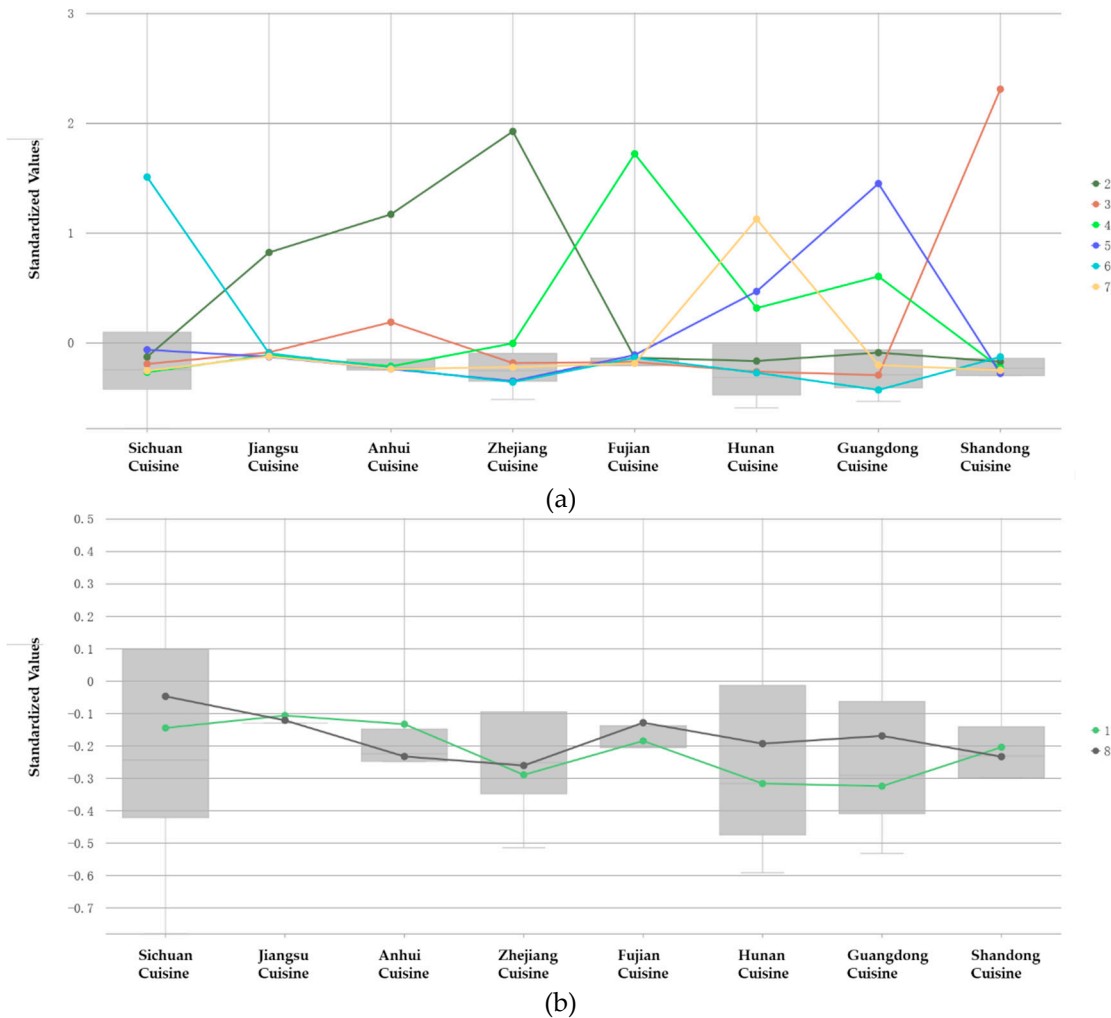

**Figure 6.** Box-plots of eight cuisines in different regions. (**a**) box-plot with high standardized values, (**b**) box-plot with low standardized values.

**Table 2.** Characteristic variables of the box-plot of eight famous Chinese cuisines.

| Cuisine Categories | Minimum | First Quartile | Median | Third Quartile | Maximum | IQR |
|---|---|---|---|---|---|---|
| Sichuan cuisine | 0 | 0.00867 | 0.01296 | 0.02122 | 0.33333 | 0.01255 |
| Jiangsu cuisine | 0.00042 | 0.00042 | 0.00042 | 0.00042 | 0.04293 | 0 |
| Anhui cuisine | 0.00088 | 0.00088 | 0.00095 | 0.0012 | 0.05375 | 0.00033 |
| Zhejiang cuisine | 0.00052 | 0.00163 | 0.00228 | 0.00333 | 0.05617 | 0.0017 |
| Fujian cuisine | 0.00086 | 0.00086 | 0.00086 | 0.00103 | 0.02713 | 0.00018 |
| Hunan cuisine | −0.00033 | −0.00025 | −0.00013 | 0.00008 | 0.00515 | 0.00033 |
| Guangdong cuisine | −0.00022 | −0.00014 | −0.00006 | 0.00009 | 0.00489 | 0.00023 |
| Shandong cuisine | 0 | 0 | 0.00009 | 0.00022 | 0.01111 | 0.00022 |

## 5. Conclusions and Future Direction

The study of food cultures from the perspective of geography emphasizes the human–environment relationship and the law of regional differentiation reflected by food cultures. Various characteristic cuisines are the carrier of local food cultures and the symbol of regional food cultures. The formation of various cuisines (such as the eight major cuisines in this study) is the objective presentation of regional differences in food cultures. Starting from the research paradigm of cultural geography, this study analyzes the spatial diffusion trend and regional differentiation pattern of food culture reflected by the eight most distinc-

tive cuisines in China from the perspectives of quantitative and regional characteristics. The spread of the eight traditional Chinese cuisines is primarily manifested in two aspects; that is, spatial distribution and spatial hierarchy. Regional differentiation is primarily reflected in the formation of food culture regions and the emergence of new cultural boundaries.

From the perspective of the quantitative characteristics of these cuisines, the farther south the regional cuisines are, the greater the spatial diffusion scope of cuisine food culture is in terms of spatial distribution. By contrast, the farther north the cuisines are, the more limited the scope is. Among these cuisines, Sichuan cuisine presents the most vigorous space diffusion, whereas Shandong cuisine presents the weakest space diffusion. For most types of cuisines, urban agglomerations, such as the Yangtze River Delta, the Pearl River Delta, and the Beijing–Tianjin–Hebei Regions, do not form the hotspots of absolute superiority due to their population aggregation, developed economy, and large population mobility. Instead, these agglomerations show a trend of outward diffusion from the origin of each cuisine. In terms of hierarchical structure, the hotspots of different levels of Shandong, Jiangsu, and Anhui cuisines are uneven, and polarization in these hotspots is evident. Other southern cuisines have a relatively balanced hotspots distribution in five grades. To sum up, regardless of spatial distribution or hierarchical structure, the overall trend of north–south differentiation is shown. The former is reflected in the diffusion range, whereas the latter is reflected in the equalization of hotspots at different levels.

This study uses prefecture-level cities as the basic research unit and takes the proportion of each cuisine as the variable to make comprehensive regionalization. The geographical regionalization results of the preference of the eight Chinese cuisines are actually the construction process of food culture regions. Each food culture region not only maintains the spatial continuity but also ensures the similarity of people's preferences for these cuisines in the cultural region. Cultural districts ensure the diversity of their preferences for these cuisines. Cultural regions and their boundaries are in dynamic change, deeply influenced by the daily production and reproduction of cuisine culture. This study constructs new cultural regions using the restaurant data for the understanding of the characteristics of food culture in different food culture regions. This study also suggests that cultural boundaries have not been disappeared because of the development of modern transportation technology and the Internet, which give a reverse response to the concern on geographical boundaries will be gone according to "The Exaggerated Death of Geography". In addition, the cultural boundary is a core issue in the field of cultural geography. This paper also provides a feasible way of thinking for the cultural boundary of quantitative extraction.

Although this paper has analyzed the spatial distribution pattern and regional structure characteristics of Chinese food culture based on restaurant POI data, there is still some work that needs to be further involved. The target of quantitative analysis of spatial structure is mechanism analysis. In future, on the one hand, we will further study how the culture region structure formed, and main factors which affected. On the other hand, only one year's restaurant data is used to regionalization in this paper, the future study will focus on different years dynamic change characteristics of the regional structure, though there is still a challenge on the acquisition of restaurant POI data for long time series in the whole of China.

**Author Contributions:** Conceptualization, Haiping Zhang and Shangjing Jiang; methodology, Haiping Zhang; software, Haoran Wang; data curation, Lei Zhou.; writing—original draft preparation, Haiping Zhang and Shangjing Jiang; writing—review and editing, Lei Zhou and Shangjing Jiang; visualization, Haoran Wang and Haiping Zhang; supervision, Guoan Tang; project administration, Guoan Tang; funding acquisition, Guoan Tang. All authors have read and agreed to the published version of the manuscript.

**Funding:** This research was funded by National Natural Science Foundation of China, grant number 41930102, 42071212and 41701185.

**Institutional Review Board Statement:** Not applicable.

**Informed Consent Statement:** Not applicable.

**Data Availability Statement:** Data sharing not applicable.

**Conflicts of Interest:** The authors declare no conflict of interest.

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
