# Peer review of "Using Restaurant POI Data to Explore Regional Structure of Food Culture Based on Cuisine Preference"

_ijgi, doi:10.3390/ijgi10010038_

Round 1

Reviewer 1 Report

In my opinion, the article is very interesting. The research methods and the results of the analysis are described in detail. The introduction provides a valuable background for further research.

 I have only two little tips:

  1. There is no information (in the last paragraph of the introduction, in chapter 2) about the period of time (year or years) the survey concerns.
  2. In table 1, two columns are named Restaurants number, but according to the text above, the last column presents percentage of restaurants in each cuisine (?) – in my opinion, something is wrong here.

Author Response

Response to Reviewer 1 Comments

In my opinion, the article is very interesting. The research methods and the results of the analysis are described in detail. The introduction provides a valuable background for further research.

 I have only two little tips:

There is no information (in the last paragraph of the introduction, in chapter 2) about the period of time (year or years) the survey concerns.

In table 1, two columns are named Restaurants number, but according to the text above, the last column presents percentage of restaurants in each cuisine (?) – in my opinion, something is wrong here.

We appreciate the time and effort that you dedicated to providing feedback on our manuscript and are grateful for the insightful comments and valuable improvements to our paper. We have incorporated most of the suggestions. Those changes are highlighted within the manuscript. Please see below, in red, for a point-by-point response to your comments and concerns. All line numbers refer to the revised manuscript file with tracked changes.

Point 1: There is no information (in the last paragraph of the introduction, in chapter 2) about the period of time (year or years) the survey concerns.

Response 1: Thanks for your reminder. The study data is collected in 2018, and the description of time information has been added in chapter 2.

See line 151.

Point 2: In table 1, two columns are named Restaurants number, but according to the text above, the last column presents percentage of restaurants in each cuisine (?) – in my opinion, something is wrong here.

Response 2: Thanks for your reminder. It’s really an error, and the error in Table 1 has been corrected, the last two columns respectively present the total number of restaurants in each cuisine and their proportion.

See line 158.

Reviewer 2 Report

This is an interesting paper, but there are a few issues which need attention.

First, it is primarily descriptive. It needs more conceptual research questions. What is the main academic literature that this paper draws from and what is the theoretical contribution of the paper? At this point, it seems like the methods and interesting graphics are driving the paper, but why do the results matter and how does it contribute to an existing theoretical research gap? A more well-developed literature review needs to be written with a clearer sense of the research questions/research gap. This is an issue both in the introduction/literature review and the discussion/conclusion at the end of the paper. 

Also, it is not clear how the methods used can be assumed to cause or produce a unique cultural landscape. I think the use of culture in this paper is not well-developed as it has a variety of meanings conceptually in the literature in cultural geography and related fields. What are the causal factors contributing to or producing such a unique landscape? In addition, the use of geography is used in an equally uncritical way in this paper as it may make more sense to speak of spatial concentrations rather than geography as it is (like culture) a contested term with multiple meanings. Additionally, how can we assume that the existence or concentration of particular types of restaurants imply a meaningful association? Correlation or concentration does not equal causation or a meaningful cluster necessarily.

I would suggest that the author work through the two broad concerns noted above and resubmit the paper. Overall, I found the paper interesting from a methods standpoint with good maps, but the conceptual framework is underdeveloped and not ready for publication at this point until it is more well-developed. The author needs to take this paper from a descriptive report-like account to a more theoretically engaged analytical paper with an emphasis on the existing literature and the relationship between the study's data and results and how these aspects impact conceptual understandings of the topic beyond the empirical aspects specific to the case study itself (the paper needs to speak more broadly to the literature).

Lastly, there are some issues with writing and typos which need to be addressed.

Good luck with the revisions.

Author Response

Response to Reviewer 2 Comments

This is an interesting paper, but there are a few issues which need attention.

First, it is primarily descriptive. It needs more conceptual research questions. What is the main academic literature that this paper draws from and what is the theoretical contribution of the paper? At this point, it seems like the methods and interesting graphics are driving the paper, but why do the results matter and how does it contribute to an existing theoretical research gap? A more well-developed literature review needs to be written with a clearer sense of the research questions/research gap. This is an issue both in the introduction/literature review and the discussion/conclusion at the end of the paper.

Also, it is not clear how the methods used can be assumed to cause or produce a unique cultural landscape. I think the use of culture in this paper is not well-developed as it has a variety of meanings conceptually in the literature in cultural geography and related fields. What are the causal factors contributing to or producing such a unique landscape? In addition, the use of geography is used in an equally uncritical way in this paper as it may make more sense to speak of spatial concentrations rather than geography as it is (like culture) a contested term with multiple meanings. Additionally, how can we assume that the existence or concentration of particular types of restaurants imply a meaningful association? Correlation or concentration does not equal causation or a meaningful cluster necessarily.

I would suggest that the author work through the two broad concerns noted above and resubmit the paper. Overall, I found the paper interesting from a methods standpoint with good maps, but the conceptual framework is underdeveloped and not ready for publication at this point until it is more well-developed. The author needs to take this paper from a descriptive report-like account to a more theoretically engaged analytical paper with an emphasis on the existing literature and the relationship between the study's data and results and how these aspects impact conceptual understandings of the topic beyond the empirical aspects specific to the case study itself (the paper needs to speak more broadly to the literature).

Lastly, there are some issues with writing and typos which need to be addressed.

Good luck with the revisions.

We appreciate the time and effort that you dedicated to providing feedback on our manuscript and are grateful for the insightful comments and valuable improvements to our paper. We have incorporated most of the suggestions. Those changes are highlighted within the manuscript. Please see below, in red, for a point-by-point response to your comments and concerns. All line numbers refer to the revised manuscript file with tracked changes.

Point 1: First, it is primarily descriptive. It needs more conceptual research questions. What is the main academic literature that this paper draws from and what is the theoretical contribution of the paper? At this point, it seems like the methods and interesting graphics are driving the paper, but why do the results matter and how does it contribute to an existing theoretical research gap? A more well-developed literature review needs to be written with a clearer sense of the research questions/research gap. This is an issue both in the introduction/literature review and the discussion/conclusion at the end of the paper.

Response 1: Thanks for your reminder and suggestions. Regionalization is the main method of describing the structural characteristics of specific geographical things or phenomena in geography. In cultural geography, there are five main themes including cultural region, cultural landscape, cultural hearth, cultural diffusion and cultural ecology. Cultural regionalization is an important way to extract and understand cultural regions. This is of great significance to dig deep into homogeneity culture region and scientifically cognize regional cultural characteristics. The revised manuscript strengthens the description of the scientific gaps and theoretical contributions of this research in the introduction, literature review and conclusion respectively.

See lines 80-91,428-444.

Point 2: Also, it is not clear how the methods used can be assumed to cause or produce a unique cultural landscape. I think the use of culture in this paper is not well-developed as it has a variety of meanings conceptually in the literature in cultural geography and related fields. What are the causal factors contributing to or producing such a unique landscape? In addition, the use of geography is used in an equally uncritical way in this paper as it may make more sense to speak of spatial concentrations rather than geography as it is (like culture) a contested term with multiple meanings. Additionally, how can we assume that the existence or concentration of particular types of restaurants imply a meaningful association? Correlation or concentration does not equal causation or a meaningful cluster necessarily.

Response 2: Thanks for your reminder and suggestions.

First of all, the cultural background of Chinese food is different from that of Western countries such as Europe and the United States. In China, food preferences vary greatly from region to region, and have always been seen as a cultural symbol which is used to distinguish cultural differences between people in different regions. This provides rationality for treating food preferences as a cultural phenomenon, as well as the possibility of regionalization based on food culture.

The reasons why we use geography rather than spatial concentrations are following: this paper not only pays attention to the phenomenon of high spatial concentrations (hotspot detection), but also pays attention to the proportion of different cuisines in the regions (food preference-based regionalization). Thus, it couldn’t be treated as spatial concentration, and the use of geography is reasonable.

As an old Chinese saying goes, "food is the most important thing for the people." People view food as the primary need. The majority of Chinese people's diet is mainly Chinese food. Although Chinese food is made up of various regional cuisines, it is not just the eight major cuisines. After a long period of evolution and its own system, the eight major cuisines with distinctive local flavoured characteristics, are widely recognized by the society and the most influential local cuisines in China. By contrast, the spread of other local cuisines is not as wide as that of the eight major cuisines, and their audience is still mainly local people in the birthplace. The audiences of the eight major cuisines are all over the country. From the perspective of cultural influence, they can be regarded as the representation of food preference. The concentration of restaurants with eight major cuisines indicates the concentration of their audiences, and the proportion of different cuisines in the analysis unit indicates the composition of food preferences in the analysis unit. Thus, Clustering and regionalization based on eight major cuisines can represent the regional structure of food culture preference.

Point 3: I would suggest that the author work through the two broad concerns noted above and resubmit the paper. Overall, I found the paper interesting from a methods standpoint with good maps, but the conceptual framework is underdeveloped and not ready for publication at this point until it is more well-developed. The author needs to take this paper from a descriptive report-like account to a more theoretically engaged analytical paper with an emphasis on the existing literature and the relationship between the study's data and results and how these aspects impact conceptual understandings of the topic beyond the empirical aspects specific to the case study itself (the paper needs to speak more broadly to the literature)

Response 3: Thanks for your reminder and suggestions. By full considerations of Point 1 and 2, we have made further improvements in the literature review and conclusion regarding your suggestion to develop the conceptual framework. This study also provides insight to the extraction and understanding of cultural boundary. More theoretical interpretation has been added in the revised manuscript.

See lines 80-91,428-444.

Point 4: Lastly, there are some issues with writing and typos which need to be addressed.

Response 4: Thanks for your reminder and suggestions. We corrected the grammar and spelling issues by reading the manuscript again and again. Such as, the wrong name of columns in table 1 was corrected.

See line 158.

Reviewer 3 Report

The article Using restaurant POI data to explore regional structure of food culture based on cuisine preference deals with the distribution of the eight most famous cuisines in China in an interesting way. At a methodological level, it is committed to the use of hot spots and grouping analyzes. However, it makes an interesting contribution when it comes to regionalizing. This regionalization is observed in the results. It takes prefecture as the unit of analysis, justifying it because with smaller units of analysis some data would be 0. This aspect could be better explained, since it is not clear to the reader. I recommend making a small comparison with units lower than the prefecture to see if the proposed method improves them. Is it possible that it is acceptable in the research that some units of analysis do not have data? Some publications affect a similar fact and choose to lose that basic information.
For this reason, I believe that the article can be published with some modifications, although it would be advisable to include a discussion about the proposed methodology and the results obtained. The methodology could include all the information related to the configuration that has been used to apply geostatistical techniques (conceptualization of distance, for example). This aspect is important because the results vary depending on the type of distance conceptualization applied. I also think it necessary to carry out a more in-depth literature review on the techniques used. In fact, only in ISPRS there are 19 publications that deal with "hot spots" and many have not been consulted. Some explain the problems inherent in their application.
Good luck

Author Response

Response to Reviewer 3 Comments

The article Using restaurant POI data to explore regional structure of food culture based on cuisine preference deals with the distribution of the eight most famous cuisines in China in an interesting way. At a methodological level, it is committed to the use of hot spots and grouping analyzes. However, it makes an interesting contribution when it comes to regionalizing. This regionalization is observed in the results. It takes prefecture as the unit of analysis, justifying it because with smaller units of analysis some data would be 0. This aspect could be better explained, since it is not clear to the reader. I recommend making a small comparison with units lower than the prefecture to see if the proposed method improves them. Is it possible that it is acceptable in the research that some units of analysis do not have data? Some publications affect a similar fact and choose to lose that basic information.

For this reason, I believe that the article can be published with some modifications, although it would be advisable to include a discussion about the proposed methodology and the results obtained. The methodology could include all the information related to the configuration that has been used to apply geostatistical techniques (conceptualization of distance, for example). This aspect is important because the results vary depending on the type of distance conceptualization applied. I also think it necessary to carry out a more in-depth literature review on the techniques used. In fact, only in ISPRS there are 19 publications that deal with "hot spots" and many have not been consulted. Some explain the problems inherent in their application.

Good luck

We appreciate the time and effort that you dedicated to providing feedback on our manuscript and are grateful for the insightful comments and valuable improvements to our paper. We have incorporated most of the suggestions. Those changes are highlighted within the manuscript. Please see below, in red, for a point-by-point response to your comments and concerns. All line numbers refer to the revised manuscript file with tracked changes.

Point 1: It takes prefecture as the unit of analysis, justifying it because with smaller units of analysis some data would be 0. This aspect could be better explained, since it is not clear to the reader. I recommend making a small comparison with units lower than the prefecture to see if the proposed method improves them. Is it possible that it is acceptable in the research that some units of analysis do not have data? Some publications affect a similar fact and choose to lose that basic information.

Response 1: Thanks for your reminder and suggestions. In China, prefectures are a relatively large geographical area. A smaller administrative unit is county. If take county-level unit as analysis unit, some units wouldn’t have all eight cuisines of restaurants (such as economically underdeveloped counties or remote areas). We tried to use the county as the analysis unit, in fact, the regionalization effect is not as good as the prefecture level. Because there are 2844 county-level units in the country, and the total numbers of restaurants of Shandong, Jiangsu and Fujian cuisine in the country do not exceed 5000. In that case, many county-level units would obtain the proportion values very close to zero or even zero in account for Shandong, Jiangsu and Fujian cuisine restaurants. The proportion of zero means the lack of regionalization indicator. It is not good for regionalization analysis. Therefore, we chose the prefecture as the analysis unit. Better explanations of data 0 have been added to the newly submitted manuscript. The only locations without data are Hong Kong, Macau and Taiwan, which we identify as the closest regionalization or classification, to some extent in line with both reality and the basic principles of regionalization.

See lines 138-142, 329-331.

Point 2: For this reason, I believe that the article can be published with some modifications, although it would be advisable to include a discussion about the proposed methodology and the results obtained. The methodology could include all the information related to the configuration that has been used to apply geostatistical techniques (conceptualization of distance, for example). This aspect is important because the results vary depending on the type of distance conceptualization applied. I also think it necessary to carry out a more in-depth literature review on the techniques used. In fact, only in ISPRS there are 19 publications that deal with "hot spots" and many have not been consulted. Some explain the problems inherent in their application.

Response 2: Thanks for your reminder and suggestions. In this study, the method for analysis of the spatial quantitative characteristics of restaurant POI (density-based hotspot detection) is proposed, but the method of regionalization based on machine learning is not originally proposed in this paper. The density-based hotspot detection method is mainly influenced by the parameters of kernel density estimation, and kernel density estimation is a mature and well-known method. Therefore, the use of parameters is not illustrated in detail. But to make it easier for readers to understand, we've added instructions in the hotspot detection Methods section. In addition, in the introduction, we supplement the literature review on hotspot detection.

See lines 108-118,189-195.

Round 2

Reviewer 2 Report

The author has worked hard to revise the manuscript in line with the reviewers' comments. The paper is stronger now. However, a few additional improvements may be necessary before publication.

First, the introduction could incorporate some of the language the author describes in their cover letter into the actual manuscript. In my opinion, when the author states in the cover letter, "Regionalization is the main method of describing the structural characteristics of specific geographical things or phenomena in geography. In cultural geography, there are five main themes including cultural region, cultural landscape, cultural hearth, cultural diffusion and cultural ecology. Cultural regionalization is an important way to extract and understand cultural regions. This is of great significance to dig deep into homogeneity culture region and scientifically cognize regional cultural characteristics. The revised manuscript strengthens the description of the scientific gaps and theoretical contributions of this research in the introduction, literature review and conclusion respectively," a version of this statement could be inserted into the abstract and intro in slightly different wordage which might help explain what the author is trying to do in the paper.

Second, in regards to the methods, I think there is also some additional language and context that the author provides in the cover letter which might be useful in the paper itself. A version of the statement below could be inserted into the intro or methods in slightly different wordage which might help explain what the author is trying to do in the paper. This paragraph provides important context for the reader.

As stated in the author's cover letter, "First of all, the cultural background of Chinese food is different from that of Western countries such as Europe and the United States. In China, food preferences vary greatly from region to region, and have always been seen as a cultural symbol which is used to distinguish cultural differences between people in different regions. This provides rationality for treating food preferences as a cultural phenomenon, as well as the possibility of regionalization based on food culture. The reasons why we use geography rather than spatial concentrations are following: this paper not only pays attention to the phenomenon of high spatial concentrations (hotspot detection), but also pays attention to the proportion of different cuisines in the regions (food preference based regionalization). Thus, it couldn’t be treated as spatial concentration, and the use of geography is reasonable...The majority of Chinese people's diet is mainly Chinese food. Although Chinese food is made up of various regional cuisines, it is not just the eight major cuisines. After a long period of evolution and its own system, the eight major cuisines with distinctive local flavoured characteristics, are widely recognized by the society and the most influential local cuisines in China. By contrast, the spread of other local cuisines is not as wide as that of the eight major cuisines, and their audience is still mainly local people in the birthplace. The audiences of the eight major cuisines are all over the country. From the perspective of cultural influence, they can be regarded as the representation of food preference. The concentration of restaurants with eight major cuisines indicates the concentration of their audiences, and the proportion of different cuisines in the analysis unit indicates the composition of food preferences in the analysis unit. Thus, Clustering and regionalization based on eight major cuisines can represent the regional structure of food culture preference." Again, I would paraphrase this language and find a way to insert the key parts of it into the paper as it provides important context for the reader.

Third, the new section called Shortcoming and Future Directions is a bit short. I wonder if it should be folded into the previous section.

Fourth, I would suggest the authors conduct a final read through before they submit it for publication to smooth out the language. Some of the paragraphs are too long as well.

Good luck!

Author Response

Response to Reviewer 2 Comments

The author has worked hard to revise the manuscript in line with the reviewers' comments. The paper is stronger now. However, a few additional improvements may be necessary before publication.

First, the introduction could incorporate some of the language the author describes in their cover letter into the actual manuscript. In my opinion, when the author states in the cover letter, "Regionalization is the main method of describing the structural characteristics of specific geographical things or phenomena in geography. In cultural geography, there are five main themes including cultural region, cultural landscape, cultural hearth, cultural diffusion and cultural ecology. Cultural regionalization is an important way to extract and understand cultural regions. This is of great significance to dig deep into homogeneity culture region and scientifically cognize regional cultural characteristics. The revised manuscript strengthens the description of the scientific gaps and theoretical contributions of this research in the introduction, literature review and conclusion respectively," a version of this statement could be inserted into the abstract and intro in slightly different wordage which might help explain what the author is trying to do in the paper.

Second, in regards to the methods, I think there is also some additional language and context that the author provides in the cover letter which might be useful in the paper itself. A version of the statement below could be inserted into the intro or methods in slightly different wordage which might help explain what the author is trying to do in the paper. This paragraph provides important context for the reader.

As stated in the author's cover letter, "First of all, the cultural background of Chinese food is different from that of Western countries such as Europe and the United States. In China, food preferences vary greatly from region to region, and have always been seen as a cultural symbol which is used to distinguish cultural differences between people in different regions. This provides rationality for treating food preferences as a cultural phenomenon, as well as the possibility of regionalization based on food culture. The reasons why we use geography rather than spatial concentrations are following: this paper not only pays attention to the phenomenon of high spatial concentrations (hotspot detection), but also pays attention to the proportion of different cuisines in the regions (food preference based regionalization). Thus, it couldn’t be treated as spatial concentration, and the use of geography is reasonable...The majority of Chinese people's diet is mainly Chinese food. Although Chinese food is made up of various regional cuisines, it is not just the eight major cuisines. After a long period of evolution and its own system, the eight major cuisines with distinctive local flavoured characteristics, are widely recognized by the society and the most influential local cuisines in China. By contrast, the spread of other local cuisines is not as wide as that of the eight major cuisines, and their audience is still mainly local people in the birthplace. The audiences of the eight major cuisines are all over the country. From the perspective of cultural influence, they can be regarded as the representation of food preference. The concentration of restaurants with eight major cuisines indicates the concentration of their audiences, and the proportion of different cuisines in the analysis unit indicates the composition of food preferences in the analysis unit. Thus, Clustering and regionalization based on eight major cuisines can represent the regional structure of food culture preference." Again, I would paraphrase this language and find a way to insert the key parts of it into the paper as it provides important context for the reader.

Third, the new section called Shortcoming and Future Directions is a bit short. I wonder if it should be folded into the previous section.

Fourth, I would suggest the authors conduct a final read through before they submit it for publication to smooth out the language. Some of the paragraphs are too long as well.

Good luck!

We are very grateful to your comments for the manuscript. According to your advice, we amended the relevant part in the newly submitted manuscript. Those changes are highlighted within the manuscript. Please see below, in red, for a point-by-point response to your comments and concerns. All line numbers refer to the revised manuscript file with tracked changes.

Point 1: First, the introduction could incorporate some of the language the author describes in their cover letter into the actual manuscript. In my opinion, when the author states in the cover letter, "Regionalization is the main method of describing the structural characteristics of specific geographical things or phenomena in geography. In cultural geography, there are five main themes including cultural region, cultural landscape, cultural hearth, cultural diffusion and cultural ecology. Cultural regionalization is an important way to extract and understand cultural regions. This is of great significance to dig deep into homogeneity culture region and scientifically cognize regional cultural characteristics. The revised manuscript strengthens the description of the scientific gaps and theoretical contributions of this research in the introduction, literature review and conclusion respectively," a version of this statement could be inserted into the abstract and intro in slightly different wordage which might help explain what the author is trying to do in the paper.

Response 1: Thanks for your reminder and suggestions. We have made the statement you mentioned above into slightly different wordage and inserted it into abstract and introduction.

See lines 7-9, 85-88.

Point 2: Second, in regards to the methods, I think there is also some additional language and context that the author provides in the cover letter which might be useful in the paper itself. A version of the statement below could be inserted into the intro or methods in slightly different wordage which might help explain what the author is trying to do in the paper. This paragraph provides important context for the reader.

As stated in the author's cover letter, "First of all, the cultural background of Chinese food is different from that of Western countries such as Europe and the United States. In China, food preferences vary greatly from region to region, and have always been seen as a cultural symbol which is used to distinguish cultural differences between people in different regions. This provides rationality for treating food preferences as a cultural phenomenon, as well as the possibility of regionalization based on food culture. The reasons why we use geography rather than spatial concentrations are following: this paper not only pays attention to the phenomenon of high spatial concentrations (hotspot detection), but also pays attention to the proportion of different cuisines in the regions (food preference based regionalization). Thus, it couldn’t be treated as spatial concentration, and the use of geography is reasonable...The majority of Chinese people's diet is mainly Chinese food. Although Chinese food is made up of various regional cuisines, it is not just the eight major cuisines. After a long period of evolution and its own system, the eight major cuisines with distinctive local flavoured characteristics, are widely recognized by the society and the most influential local cuisines in China. By contrast, the spread of other local cuisines is not as wide as that of the eight major cuisines, and their audience is still mainly local people in the birthplace. The audiences of the eight major cuisines are all over the country. From the perspective of cultural influence, they can be regarded as the representation of food preference. The concentration of restaurants with eight major cuisines indicates the concentration of their audiences, and the proportion of different cuisines in the analysis unit indicates the composition of food preferences in the analysis unit. Thus, Clustering and regionalization based on eight major cuisines can represent the regional structure of food culture preference." Again, I would paraphrase this language and find a way to insert the key parts of it into the paper as it provides important context for the reader.

Response 2: Thanks for your reminder and suggestions. The important information about the context of the research methods has been added to the newly submitted manuscript.

See lines 137-150.

Point 3: Third, the new section called Shortcoming and Future Directions is a bit short. I wonder if it should be folded into the previous section.

Response 3: Thanks for your reminder and suggestion. We have adopted your suggestion and folded it into the previous section.

See lines 455-462.

Point 4: Fourth, I would suggest the authors conduct a final read through before they submit it for publication to smooth out the language. Some of the paragraphs are too long as well.

Response 4: Thanks for your reminder and suggestion. We have reread the manuscript and made some long paragraphs shorter by subsection. Such as, the second paragraph and the second and seventh paragraph in Introduction

See lines 56,115.

Reviewer 3 Report

I think the article has improved on the most important points. Now the reader understands both the units of analysis used and the proposed methodology.
I have only detected that reference 48 and 49 are identical.

good luck

Author Response

Response to Reviewer 3 Comments

I think the article has improved on the most important points. Now the reader understands both the units of analysis used and the proposed methodology.

I have only detected that reference 48 and 49 are identical.

good luck

We are very grateful to your comments for the manuscript. According to your advice, we amended the relevant part in the newly submitted manuscript. Those changes are highlighted within the manuscript. Please see below, in red, for a point-by-point response to your comments and concerns. All line numbers refer to the revised manuscript file with tracked changes.

Point 1: I have only detected that reference 48 and 49 are identical.

Response 1: Thanks for your reminder.  We have made corrections to the reference and deleted the identical reference.

See lines 567-568.
